# Hybrid Fly Ash-Based Geopolymeric Foams: Microstructural, Thermal and Mechanical Properties

**DOI:** 10.3390/ma13132919

**Published:** 2020-06-29

**Authors:** Giuseppina Roviello, Laura Ricciotti, Antonio Jacopo Molino, Costantino Menna, Claudio Ferone, Domenico Asprone, Raffaele Cioffi, Veronica Ferrandiz-Mas, Pietro Russo, Oreste Tarallo

**Affiliations:** 1Department of Engineering, University of Naples ‘Parthenope’, Centro Direzionale, Isola C4, 80143 Napoli, Italy; giuseppina.roviello@uniparthenope.it (G.R.); antoniojacopo.molino@uniparthenope.it (A.J.M.); claudio.ferone@uniparthenope.it (C.F.); raffaele.cioffi@uniparthenope.it (R.C.); 2INSTM Research Group Napoli Parthenope, National Consortium for Science and Technology of Materials, Via G. Giusti, 9 50121 Firenze, Italy; 3Department of Structures for Engineering and Architecture, University of Naples Federico II, via Claudio 21, 80125 Napoli, Italy; costantino.menna@unina.it (C.M.); domenico.asprone@unina.it (D.A.); 4Department of Architecture & Civil Engineering BRE Centre in Innovative Construction Materials (BRE CICM), University of Bath, Bath 04530, UK; vfm24@bath.ac.uk; 5Institute for Polymers, Composites and Biomaterials, National Research of Council, Via Campi Flegrei 34, 80078 Pozzuoli (NA), Italy; pietro.russo@unina.it; 6Department of Chemical Sciences, Università degli Studi di Napoli Federico II, Complesso Universitario di Monte S. Angelo, via Cintia, 80126 Napoli, Italy; oreste.tarallo@unina.it

**Keywords:** hybrid foams, lightweight material, geopolymer, fly ash, polysiloxanes, microtomography, thermal conductivity, mechanical properties

## Abstract

This research investigates the preparation and characterization of new organic–inorganic geopolymeric foams obtained by simultaneously reacting coal fly ash and an alkali silicate solution with polysiloxane oligomers. Foaming was realized in situ using Si^0^ as a blowing agent. Samples with density ranging from 0.3 to 0.7 g/cm^3^ that show good mechanical properties (with compressive strength up to ≈5 MPa for a density of 0.7 g/cm^3^) along with thermal performances (λ = 0.145 ± 0.001 W/m·K for the foamed sample with density 0.330 g/cm^3^) comparable to commercial lightweight materials used in the field of thermal insulation were prepared. Since these foams were obtained by valorizing waste byproducts, they could be considered as low environmental impact materials and, hence, with promising perspectives towards the circular economy.

## 1. Introduction

Over the past decades, considerable progress has been made in the field of energy efficiency by reducing energy consumption through rational use of energy and the development of new thermal insulating materials. These materials are able to limit heat exchange between the interior and exterior of buildings, thus improving the quality of life of their occupants and offering both environmental and economic advantages [1].

Commercially available systems for thermal insulation can be roughly classified as: (i) lightweight polymeric materials, such as expanded polyurethane and polystyrene; (ii) lightweight natural materials, such as wood, wool and cork and (iii) lightweight inorganic materials such as cellular cement or autoclaved aerated concrete (AAC). Notably, polymeric and natural materials are able to offer good performances in terms of thermal (and acoustic) insulation while they have poor mechanical properties and can become very dangerous in case of fire because of their high flammability and the possible release of toxic gases. On the contrary, AAC shows interesting mechanical properties as well high thermal resistance and no flammability [2]. For these properties, AAC is often considered the first choice for a safe building insulation.

Unfortunately, AAC is a cement-based system requiring the consumption of significant amounts of energy and natural resources for its production [3,4].

Moreover, AAC elements are usually cured in big autoclaves at high temperatures (up to 190 °C) and pressures (12–14 bar) for 12 h, resulting in a very expensive production process, both from an economic and environmental point of view [5,6].

For all these reasons, much attention is being currently devoted to the development of alternative lightweight materials, characterized by a lower cost and a lower environmental impact as compared to cement and concrete-based materials.

In this framework, a particularly interesting research area involves geopolymers, amorphous framework silicates obtained by a polycondensation reaction of an aluminosilicate source (e.g., metakaolin, fly or bottom ashes, blast furnace slag) and an alkali–silicate solution in alkaline aqueous environment [7,8,9].

Thanks to their interesting mechanical properties such as low shrinkage, their thermal stability, freeze–thaw resistance, chemical and fire resistance, long term durability and recyclability, geopolymers can find applications in fields related to refractory materials, binder for hazardous or radioactive waste encapsulation, high-tech ceramics and adhesive [10]. Moreover—compared to ordinary Portland cement (OPC)—geopolymers could possibly reduce CO_2_ emissions thanks to the low carbon footprint of some of the raw materials from which they can be prepared (such as industrial waste, fly ash or mud) [11,12]. However, geopolymers suffer of brittle mechanical behavior that limits their application in buildings.

In order to overcome this limitation, a great effort has been done in developing composites or organic–inorganic hybrid geopolymeric materials with improved mechanical properties with respect to unmodified ones [13,14,15,16,17,18,19,20,21,22,23,24]. To this aim, in the last years, this research group has established the synthesis of new geopolymer-based materials by means of a co-reticulation reaction between an organic component (i.e., an epoxy resin precursor or polysiloxane oligomers) and an inorganic silicate source in an aqueous alkaline medium [14,20]. In particular, organic–inorganic geopolymer-based hybrid materials have been obtained in both solid and foamed forms by reacting metakaolin and an alkali silicate solution with mixtures of dialkylsiloxane oligomers [20,21]. As far as these new lightweight materials, they are characterized by remarkable mechanical properties, good fire resistance along with low thermal conductivity, significantly better than those shown by unmodified geopolymer foams reported in the literature and comparable to AAC [21,22,23,24]. Moreover, it is worth pointing out that these properties could be widely tuned depending on reaction conditions and chemical composition of the starting slurry [20,21].

In this study, the possibility of preparing lightweight hybrid materials from industrial waste such as fly ash (instead of the more expensive metakaolin) is presented. In this way, lightweight insulating systems with lower cost and limited environmental impact have been prepared, thus providing a possible contribute towards the circular economy. In addition, hybrid foams containing also lightweight aggregates such as expanded clay particles [25] have been prepared and studied too. The obtained samples have been characterized from the point of view of their morphology, microstructure, thermal and mechanical properties.

## 2. Experimental Section

### 2.1. Materials and Methods

EFA-Füller HP fly ash was supplied by BauMineral GmbH (Herten, Germany). Its composition, obtained by X-ray fluorescence spectroscopy, is reported in Table 1. Sodium hydroxide with reagent grade, was supplied by Sigma-Aldrich (St. Louis, MI, USA). A sodium silicate solution, whose composition is reported in Table 1, was supplied by Prochin Italia Srl (Caserta, Italy). A commercial dimethylsiloxane oligomers mixture, Globasil AL20, was purchased from Globalchimica Srl (Turin, Italy). Silicon powder (∼325 mesh) was purchased from Sigma-Aldrich. Lightweight expanded clay (Leca Granulare) with grain size of 2–5 mm was supplied by Laterlite S.p.A. (Rubbiano di Solignano, Italy). Additional experimental details are reported in references 20 and 21.

### 2.2. Specimen Preparation

#### 2.2.1. Solid Geopolymers

The alkaline activating solution used for the activation of fly ash for the geopolymerization reaction was prepared by mixing the sodium silicate solution with a 12-M aqueous solution of sodium hydroxide. The solution was then allowed to equilibrate and cool for one day. The composition of the obtained alkaline activating solution can be expressed as Na_2_O 1.21 SiO_2_ 11.09 H_2_O. Fly ash was then dispersed into the activating solution with a liquid to solid ratio of 0.5:1 by weight and mixed by a mechanical mixer for 10 min at 400 rpm. The composition of the whole geopolymeric slurry can be expressed as Al_2_O_3_ 4.65 SiO_2_ 0.78 Na_2_O 7.78 H_2_O, assuming that geopolymerization occurred at 100% (see discussion in Section 3.1). This system was identified as the “unmodified” FA-based geopolymer and indicated in the following as “G”. Unmodified FA-based geopolymeric specimens containing also expanded clay (G–EC) were prepared as described for G samples, but by further adding 7.5% by weight of expanded clay particles into the paste. Mechanical mixing at 400 rpm was then performed for 10 min by means of a mechanical mixer.

#### 2.2.2. Geopolymer–Polysiloxane Hybrids

Solid hybrid polysiloxane–geopolymer samples (GSyl) were prepared by incorporating 10% by weight of a commercial mixture of dimethylsiloxane oligomers into a freshly prepared geopolymeric suspension (obtained as described in paragraph 2.2.1) under mechanical stirring. The addition was performed when the polycondensation reactions that lead to formation of both geopolymer and polydimethylsiloxane resin were already started, but far from being completed (see ref. 20 for further details). Hybrid specimens containing also expanded clay (GSyl–EC) were prepared as described above, but by adding also 7.5% by weight of expanded clay particles into fly ash. In both cases, the resulting mixture was mechanically mixed for 10 min at 400 rpm.

In order to obtain a set of foamed samples with a different degree of porosity, silicon powder was added into the geopolymeric suspension in different wt % ratios (0.06, 0.09 and 0.12 wt %); afterwards, the system was mixed for further 5 min at 400 rpm.

The consolidated geopolymeric hybrid foams, obtained through the above-mentioned procedure are hereafter indicated in the case of samples without expanded clay as GSyl*xx* (where “*xx*” refers to the decimal units of % *w/w* content of silicon added as a foaming agent to the geopolymer paste), while as GSyl–EC*xx* in the case of foamed samples containing also expanded clay (EC).

The mix design details of specimens are reported in Table 2.

#### 2.2.3. Curing Treatments

As soon as prepared, geopolymer (G), geopolymer hybrid (GSyl), geopolymer with expanded clay (G–EC) and geopolymer hybrid samples with expanded clay (GSyl–EC) were casted in cubic molds and cured in >95% relative humidity conditions at 60 °C for 24 h. Subsequently, the specimens were kept at room temperature for further 6 days in >95% relative humidity conditions and then for further 21 days in open air.

At variance, as soon as prepared, foamed hybrid geopolymers (GSyl*xx*) and the corresponding foamed hybrids with expanded clay (GSyl–EC*xx*) were casted in cubic molds and cured in >95% relative humidity conditions at room temperature (≈22 °C) for 24 h and then at 60 °C for further 24 h. Subsequently, the specimens were kept at room temperature for further 5 days in >95% relative humidity conditions and then for further 21 days in air.

### 2.3. Physical and Microstructural Assessment

The microstructure of all samples was assessed by SEM analysis performed by means of a Nova NanoSem 450 FEI Microscope. The internal structure of the samples was studied also by micro–computed tomography (µCT) using a µCT scanner (Skyscan 1272, Bruker Kontich, Belgium) at ATeN Center, University of Palermo (Italy). The samples were scanned at a source voltage and a current of 40 kV and 250 mA, respectively, with a total rotation of 180° and a rotation step of 0.2°. A 1-mm aluminum filter was chosen for the acquisitions. The image pixel size was 7.4 µm and the scan time was about 3 h for every sample. The reconstructions were carried out using the software NRecon (version 1.6.10.2) starting from the acquired projection images. The 2D-images have color depth of 8 bit with 265 levels of gray. After that, the whole set of raw images were displayed in a 3D-space by CTVox software. Quantitative analyses were carried out via CTan software (version 1.16.1.0). These measurements allowed also evaluating the porosity of the samples.

### 2.4. Apparent Density Determination

Apparent density measurements were carried out by means of an OHAUS-PA213 balance provided by Pioneer according to the following equation:(1)$$D=\frac{m_{d}}{m_{s}-m_{i}}$$
where m_*d*_ is the dry weight of the sample, m_*s*_ is the weight of the soaked sample and m_*i*_ is the weight of the soaked immersed sample. In order to determine their dry weight (m_*d*_), the samples were dried at 110 °C for 12 h and weighed after cooling at room temperature. After that, the specimens were placed in an empty desiccator and kept in vacuo for 30 min. Later, the desiccator was filled with water and the samples were kept immersed for 2 h in vacuum and then weighed in order to obtain the weight of soaked sample (m_*s*_). The weight of the soaked immersed sample (m_*i*_) was then obtained by weighting the samples immersed in water at atmosphere pressure.

### 2.5. Compressive Behavior

Uniaxial compressive tests were carried out on 50 × 50 × 50-mm^3^ cubic specimens by means of a MTS 810 servo-hydraulic universal testing machine. For each sample type, at least four specimens were tested under displacement control in order to obtain the corresponding stress–strain curve, compressive strength and Young’s modulus. The compressive tests were performed at a constant displacement velocity of 0.20 mm/min until the sample broke. The measurement of the displacement was given by the crosshead displacement while the Young’s modulus of each sample was computed from the linear stress–strain response recorded during the test. The values of these mechanical parameters reported in the text are the averages of the four performed tests.

### 2.6. Thermal Characterization

Thermal conductivity tests were performed on 50 × 50 × 50 mm^3^ cubic samples using a hot disk TPS 500 analyzer (Thermal Instrument, Ltd). This is a nondestructive test based on the transient plane source technique according to ISO 22007-2: 2015. Samples were tested without cutting, using the single-sided testing module, which needs of an insulating sample support of expanded polystyrene (EPS).

## 3. Results and Discussion

### 3.1. Sample Preparation

Lightweight fly ash-based hybrids (GSyl*xx*), with different density and porosity, were prepared by extending a synthetic method earlier developed by the authors for metakaolin-based hybrid foams [21]. Direct foaming was carried out by in situ generation of hydrogen gas through the addition of Si^0^ powders into the freshly prepared slurry [26]:Si (s)+ 4H_2_O (l) → Si(OH)_4_ (aq) + 2H_2_ (g)(2)

It is worth pointing out that by following the same experimental procedure, it was not possible to obtain “unmodified” geopolymer foams (i.e., without the addition of oligomeric dimethylsiloxanes in the geopolymer slurry) due to the very low viscosity of the suspension, that yielded a rapid collapse of the foamed structure initially formed due to the coalescence of individual cells.

On the contrary, thanks to higher viscosity of hybrid systems [21], the addition of the foaming agent to GSyl slurries produced homogeneously foamed structures (Figure 1) that gives rise to smaller cells with superior stability, in equilibrium with their surrounding matrix and uniformly distributed in the foamed material. In fact, during the expansion process, a balance is reached between the process of formation of the cells, which creates the increase in volume, and the consolidation of the system, which allows the voids to be blocked inside the structure. This therefore allows obtaining a stable lightweight material, with interesting mechanical properties as will be shown later on.

Of course, the foaming process was strongly dependent on the amount of Si^0^ added as a foaming agent to the freshly prepared GSyl slurry. As expected, the volume expansion of the slurry increased by increasing the amount of the foaming agent; as a consequence, the density of the cured porous materials decreased as the amount of foaming agent increased (Table 2). An analogous trend was obtained also for the set of hybrid geopolymer foams containing lightweight aggregates (Table 2). The apparent densities of foamed GSyl and GSyl–EC (Table 2) are reported in Figure 2 as a function of the wt % of Si powder added to the geopolymer slurry during foaming process. 

As stated before, both foamed materials exhibited a decrease of bulk density with increasing the Si^0^ content. However, data suggested that GSyl foams were characterized by lower density values than GSyl–EC foams when using an identical content of foaming agent. In particular, GSyl-foamed material exhibited an average apparent density varying between 0.33 and 0.67 g/cm^3^ whereas, in the case of GSyl–EC-foamed material, the density values varied between 0.45 and 0.68 g/cm^3^. The fact that, for the same amount of foaming agent, GSyl foams were less dense than the sample with EC could be mainly related to the presence of expanded clay particles that are not involved in the foaming process since are already set when they are mixed to the geopolymeric slurry and present a density of 0.68 g/cm^3^ [27]. On the contrary, in the case of GSyl, all the sample mass is involved in the foaming process, thus obtaining a more efficient process that turns out in a material with lower density. Moreover, the presence of expanded clay particles could act as physical obstacle for pore growth.

### 3.2. Structural Analysis

Optical photographs of the core section of the foams have been already shown in Figure 1. All samples are characterized by a macrostructure with roughly spherical macropores, deriving from the foaming process, rather uniformly distributed in the specimens. For each Si^0^ composition, only a minor fraction of pores characterized by more irregular shapes is present, probably due to the coalescence of spherical pores (as will be apparent by examining Figure 3f. An analogous morphology was obtained for the hybrid geopolymer phase also in the case of foamed samples containing expanded clays (not reported).

A more detailed evaluation of the pore structure as well as of its overall porosity and pore-size distribution, was evaluated for GSyl06 sample by X-ray microtomography. Figure 3e,f shows slice images and 3D-images of sample GSyl06 and, for comparison, those relative to solid G (Figure 3a,b) and GSyl (Figure 3c,d) specimens. In Figure 4 the corresponding pore-size distributions are reported.

In the case of dense specimens, microtomography images (Figure 3a–d) show that both systems are characterized by a uniform structure with diffuse macroporosity (G sample showed a total porosity of 7% while in the case of GSyl hybrid sample, porosity is 20%) due to either air bubbles entrapped in the slurry during the mixing step and water evaporation that took place during the curing time. Consistently, as apparent from Figure 4, this porosity is mainly represented by open pores with diameters up to 50 µm, and virtually no pores with diameter > of 300 µm are present.

At variance, in the case of the foamed sample (GSyl06, Figure 3e,f), the porosity is due to pores characterized by diameters spanning in a wider range (up to millimeters) and the total porosity turned out to be 67%. Moreover, even if GSyl06 sample have 90% of the pores with size between few micron up to 100 μm (Figure 4a), the most important contribution to porosity is represented by the pores with diameters > of 900 µm (Figure 4b). This simple consideration is in perfect agreement with microtomography images shown in Figure 3e,f. Furthermore, for this sample, the total porosity turned out to be constituted almost exclusively by open porosity while closed porosity represented less than 5%, markedly higher than that found for solid samples (for which closed porosity was around 6 × 10^−3^% for G and 5 × 10^−2^% for GSyl). Again, these data could be rationalized considering the fact that while in the case of solid samples po res derive only from air bubbles trapped during the mixing step and by water evaporation, for GSyl06, in addition to this kind of porosity (that is concentrated in the pore walls as will be apparent by examining SEM images reported below), pores are generated mainly by H_2_ expansion.

In order to get a better insight of the microstructure of the samples prepared, SEM micrographs of freshly obtained fractured surfaces were performed. Figure 5 shows SEM images of GSyl06 foam (obtained with Si^0^ content equal to 0.06 wt %, Figure 5a–d) and, for comparison, those relative to the solid fly ash-based geopolymer (G, Figure 5e,f) and for the hybrid solid sample (GSyl, Figure 5g,h).

As far as the hybrid-foamed geopolymer (Figure 5a−d), the results of SEM investigations support the findings of the microtomography, but additionally provide insight into the structure of the walls between the pores: these walls keep a rather compact structure in which it is possible to identify unreacted fly ash particles embedded in the geopolymer matrix, but, in agreement with microtomography data, they present several macropores with diameter <20 µm (Figure 5a–c).

As far as hybrid geopolymer with expanded clay, SEM micrographs reported in Figure 6 show the morphology of G–EC (Figure 6a,b) and GSyl–EC (Figure 6c,d) solid samples. In these cases, the most interesting feature to point out in this scale length is the interfacial transition zone between the geopolymer matrix and the surface of an expanded clay particle. In particular, while in the G–EC sample it is possible to observe the presence of micro-cracks between the matrix and the EC particles, in the case of hybrid GSyl sample instead, a sort of continuous phase, with no fractures, is present. This continuous morphology could be attributable to an improved compatibility between the hybrid geopolymer matrix with the EC particle used.

### 3.3. Thermal Conductivity

Figure 7 shows thermal conductivity data of GSyl and GSyl–EC foams. The values of thermal conductivity for GSyl foams are in the range 0.145–0.232 W/m K while for GSyl–EC foams are in the range 0.172–0.254 W/m K. For both type of samples, the values of thermal conductivity are strictly dependent on apparent density values. In fact, the lower the apparent density the lower the thermal conductivity is. The sample with the highest thermal conductivity of 0.254 ± 0.001 W/m·K was the GSyl–EC06. This sample, containing expanded clay and the lowest amount of silica powder as a foaming agent, has the highest bulk density of 0.681 g/cm^3^. Meanwhile, the sample with the lowest thermal conductivity of 0.145 ± 0.001 W/m·K resulted to be the GSyl12, without expanded clay and with the highest content of silica powder, conditions that led to the lowest density of 0.330 g/cm^3^.

Figure 8 reports the correlation between thermal conductivity and density of the hybrid foams. Several lightweight concretes samples, both aerated and containing lightweight fillers, were included for comparison [28]. It is worth pointing out that the examined hybrid foams show thermal performances comparable to different representative lightweight concretes traditionally used in the field of thermal insulation [28].

In particular, our samples are comparable with concrete samples containing EC with similar density such as Latermix Cem Classic [29], which is a premixed lightened product for layers of insulation and for thick substrates widely used in the Italian market. This product shows d = 0.600 g/cm^3^, λ = 0.134 W/m·K, σ = 2.5 MPa^,^ values [29] not very different from our materials. For these reasons, also for the hybrid samples presented in this study, similar fields of application could be suggested.

### 3.4. Mechanical Properties

Mechanical properties of foamed GSyl and GSyl–EC hybrid geopolymers are here discussed in terms of uniaxial compressive stress–strain behavior, including compressive strength and Young’s modulus evaluation. The stress–strain curves of foamed GSyl and GSyl–EC are reported in Figure 9A while corresponding compressive strength (maximum stress corresponding to the peak of the compression force) and strain, ultimate failure strain and elastic modulus are reported in Table 3.

All the investigated foamed materials exhibited a well-defined elastic regime, already detectable at the early stages of stress (stage 1). It is interesting to note that in the very first part of the stress–strain curves (i.e., corresponding to strains lower than 0.25%), the samples characterized by similar densities (GSyl06 and GSyl–EC06, GSyl09 and GSyl–EC09, GSyl12 and GSyl–EC12) show almost perfectly coincident curves pointing out a similar stiffness in compressive response (Table 3). The linear elastic regime remained with an almost constant slope until reaching the yield or unstable collapse point (stage 2) characterized by a load loss on the stress–strain curve, associated with the progressive collapse mechanism of the cells. However, while for GSyl samples a sudden drop of load was recorded and all the samples showed an ultimate failure strain lower than 1% (solid lines in Figure 9A, in the case of specimens with expanded clay (dotted lines in Figure 9A a greater ultimate failure strain was recorded, probably due to the presence of the EC particles that act as reinforce.

By plotting compressive strength and Young’s modulus of the foamed hybrids as a function of their apparent density (Figure 10) some important observations can be pointed out. First of all, the compressive strength variation of the two set of samples appears to be linear with respect to apparent density values. As expected, with the increase of the density of the specimens, it is possible to notice an improvement of their mechanical properties for both kinds of hybrid materials, almost independently from the presence of EC (that, actually, seems to have no significant effect on the mechanical performances of the materials, as could be argued by analyzing the stress–strain curves reported in Figure 9A and data reported in Table 3). As far as GSyl-foamed samples, for apparent density of 0.33 g/cm^3^, 0.38 g/cm^3^ and 0.67 g/cm^3^ the corresponding compressive strength values were 0.7 MPa, 1.0 MPa and 2.0 MPa, respectively. Instead, GSyl–EC samples with density of 0.45 g/cm^3^ 0.52 g/cm^3^ and 0.68 g/cm^3^ reported compressive strength values of 1.0 MPa, 1.5 MPa and 1.9 MPa, respectively. These data are comparable with those reported in literature for geopolymeric porous materials for which it is well demonstrated that compressive strength depends on their density and has been found to be between 1 and 10 MPa for densities of between 0.36 and 1.4 g/cm^3^ [30,31,32].

As far as Young’s modulus for both sets of hybrid foams (Figure 10b), one could observe an increase in the values recorded as the density of the samples increases. In particular, correspondingly to the same increase in the apparent density of the specimens, it could be noted that in the samples containing EC, Young’s modulus would seem to increase in percentage much greater than in samples without EC. This datum could be traced to the presence of the particles of EC which, being very rigid, as the material becomes denser and less fragile, contributes to increasing its elastic modulus. These differences between the two sets of samples tended to decrease for the specimens with higher density since in these samples the elastic modulus is likely to be mainly due to the geopolymer matrix that cements together the EC particles.

Finally, in view of a scale-up process for the production of the foams, mechanical performance were investigated also for a foamed GSyl sample, prepared by using the same strategy of synthesis, but making a scale-up of the production process (the same mix design and experimental conditions were used). In particular, samples with a volume of 10 dm^3^ were successfully prepared, and then cut to form cubic specimens with dimension of 50 × 50 × 50 mm^3^. In this way, lightweight GSyl06ls (where “ls” stand for “large scale production”) samples with apparent density of 0.70 g/cm^3^ (i.e., comparable to that one of GSyl06 and GSyl–EC06 samples obtained in the 125 cm^3^ molds) were prepared. These samples showed significantly better mechanical performances in respect to foams obtained in laboratory scale molds since they showed a compressive strength up to 5.3 MPa (magenta curve in Figure 9B while elastic modulus is the same of the lab scale samples. This great enhancement of the mechanical properties of this material could be rationalized considering the fact that the geometry of the mold affects the foam volume expansion and the consolidation rate, that, in turn, determine the development of small or big pores and their collapse when coalescence occurs [30,31]. In fact, the foaming process and the forming cells are strongly affected by shape and volume of the mold since the possibility to have vast space in which the cells can be homogenously distributed (Figure 11), allows to minimize coalescence phenomena and standardizing the shape and size of the voids [30,31] thus reducing the brittleness of the final foams.

Finally, it is worth pointing out that, also in this case, the curve that describes the elastic regime (in magenta, Figure 9B) was well-defined (stage 1). Moreover, the linear elastic regime remained with an almost constant slope until reaching the yield or unstable collapse point (stage 2) characterized by a load loss on the stress–strain curve. At this stage, the specimen showed a sort of pseudoplasticity, continuing to carry a stress of about 5 MPa up to relatively large strains, i.e., 7% axial deformation, likely due to the crushing of the pores. At this point, macrocracking begins to develop in the specimen and a brittle rupture of the specimen occurs.

## 4. Conclusions

Lightweight fly ash-based geopolymer hybrids were successfully prepared by simultaneous reacting fly ash and an aqueous alkali silicate solution with mixtures of dialkylsiloxane oligomers in the presence of Si^0^ as a foaming agent and cured at room temperature (≈22 °C).

These foamed hybrids (with density ranging from 0.3 to 0.7 g/cm^3^) showed homogeneous and very regular porosity, with voids uniformly distributed within the samples. This morphologic feature allowed obtaining interesting thermal and mechanical properties—comparable with those reported in literature for geopolymeric porous materials and for some commercial inorganic-foamed materials with similar densities (such as Aerated Autoclaved Concrete or concrete with lightweight aggregates). In particular, in the case of the hybrid geopolymer foam with density of ≈0.7 g/cm^3^, a compressive strength up two megapascals was recorded while, as far as thermal conductivity, λ_10_ values in the range of 0.145–0.254 W/mK were obtained for samples characterized by density in the range 0.33–0.68 g/cm^3^, respectively. In particular, the foamed sample with density 0.330 g/cm^3^ and not containing EC showed the best performance in terms of thermal insulation.

Finally—in view of a scale-up process for the production of the foams—mechanical performances were also investigated for foamed samples with density of 0.7 g/cm^3^ prepared by the same experimental procedure, but producing large volume specimens (10 dm^3^). For these materials, a significant improvement of mechanical properties was recorded, with compressive strength up 5.3 MPa, much better than those reported for some commercial inorganic-foamed materials with similar densities. This significant improvement could be due to a more uniform foaming process in respect to that obtained for the small volume samples.

For all these reasons—and considering the fact that these materials were obtained by valorizing industrial byproducts, thus limiting the construction material costs and, at the same time, reducing its environmental impact—these innovative hybrid foams could be considered valuable alternative to cement base ones for thermal insulation.

## Figures and Tables

**Figure 1 materials-13-02919-f001:**
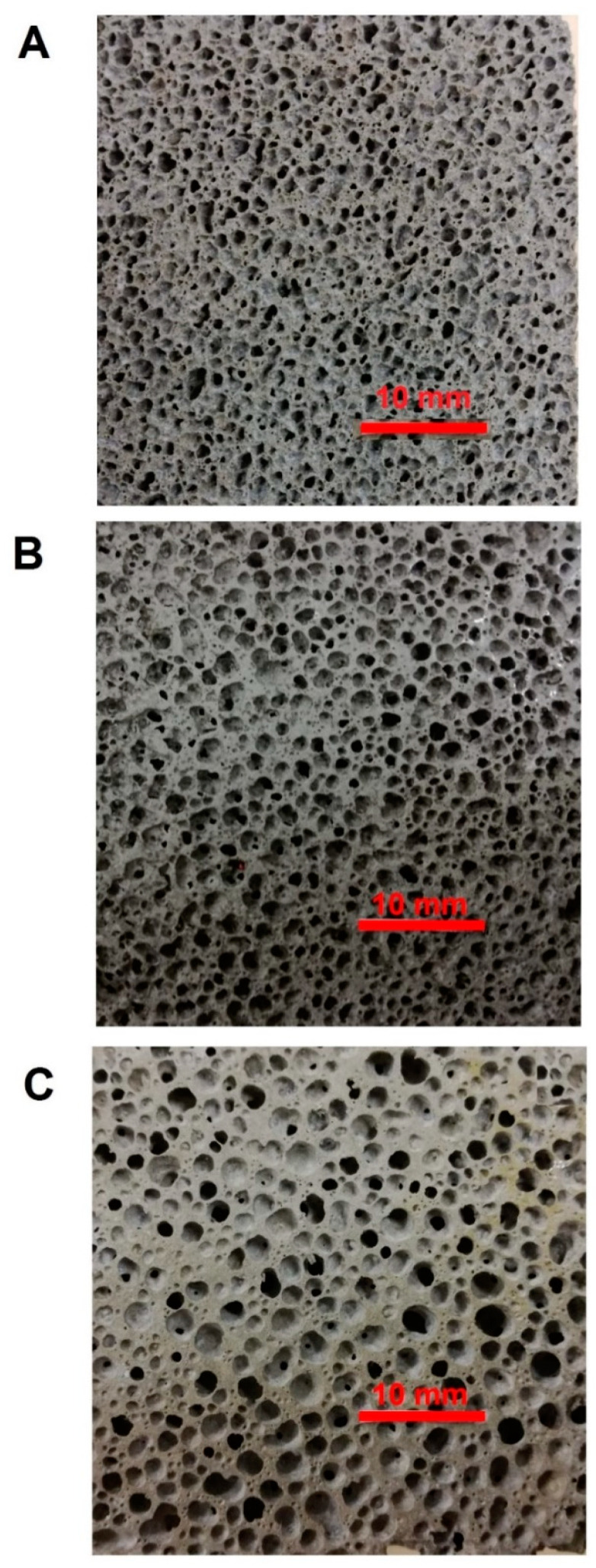
Optical images of polished section surfaces of GSyl06 (**A**), GSyl09 (**B**) and GSyl12 (**C**) prepared by adding 0.06 wt %, 0.09 wt % and 0.12 wt % respectively of Si^0^ to the hybrid geopolymeric slurries.

**Figure 2 materials-13-02919-f002:**
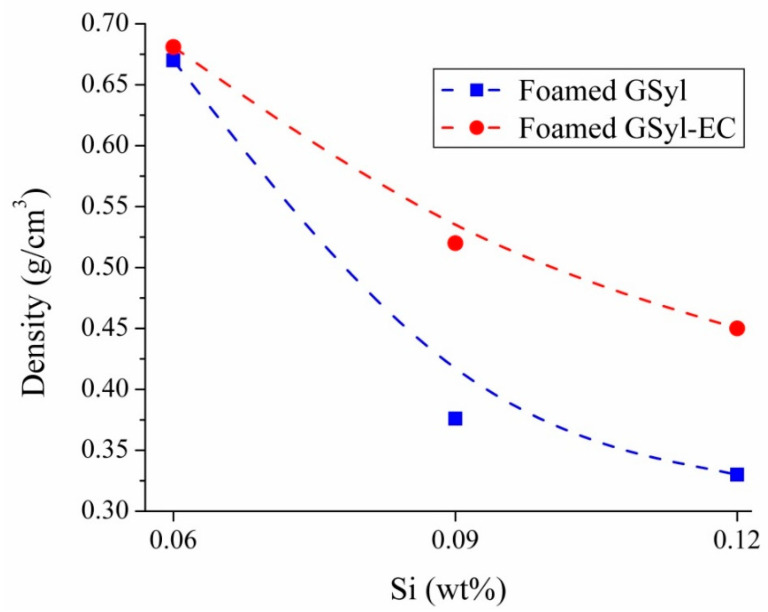
Averaged apparent density (over three samples) of foamed GSyl (blue squares) and GSyl–EC (red balls) as a function of Si^0^ content. Lines are a guide for the eye.

**Figure 3 materials-13-02919-f003:**
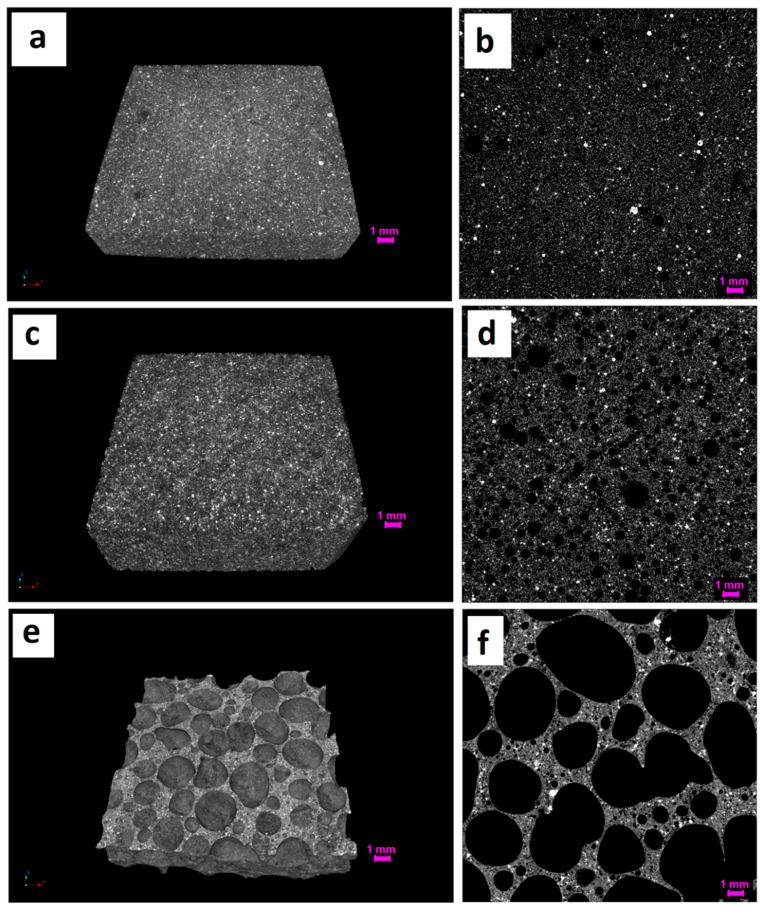
Three-dimensional (left) and 2D (right) slice images obtained by microtomography of solid unmodified geopolymer G (**a**,**b**), solid hybrid GSyl (**c**,**d**) and GSyl06 hybrid foam (**e**,**f**). Scale bar is 1 mm.

**Figure 4 materials-13-02919-f004:**
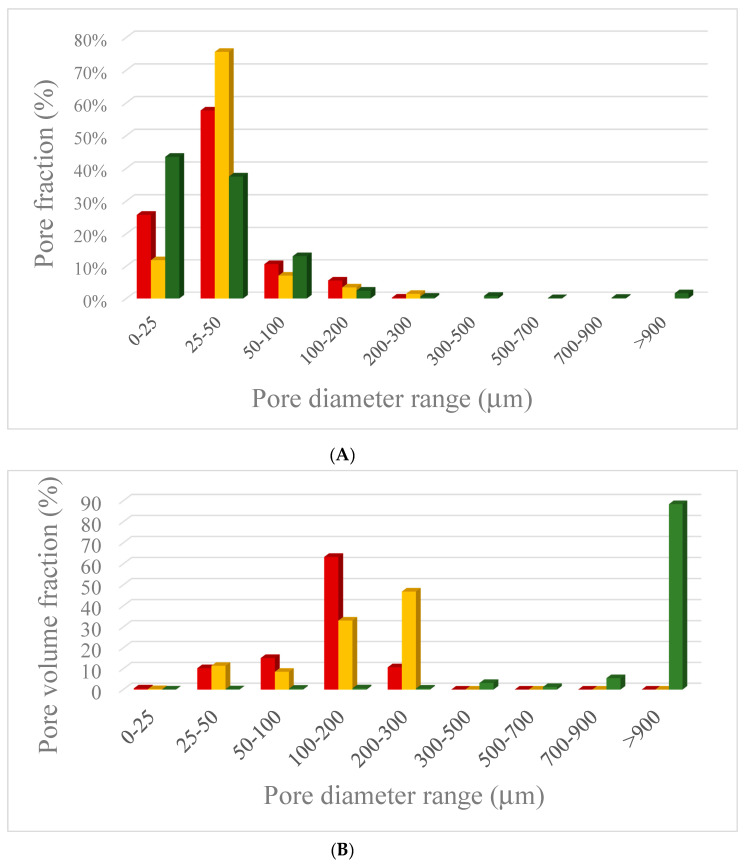
(**A**) Pore-size distribution and (**B**) pore-volume fraction for the samples G (red bars), GSyl (yellow bars) and GSyl06 (green bars) as obtained by analysis of microtomography reported in Figure 3. In B, pore-volume fractions were calculated by imposing as 100 the empty volume of each sample independently.

**Figure 5 materials-13-02919-f005:**
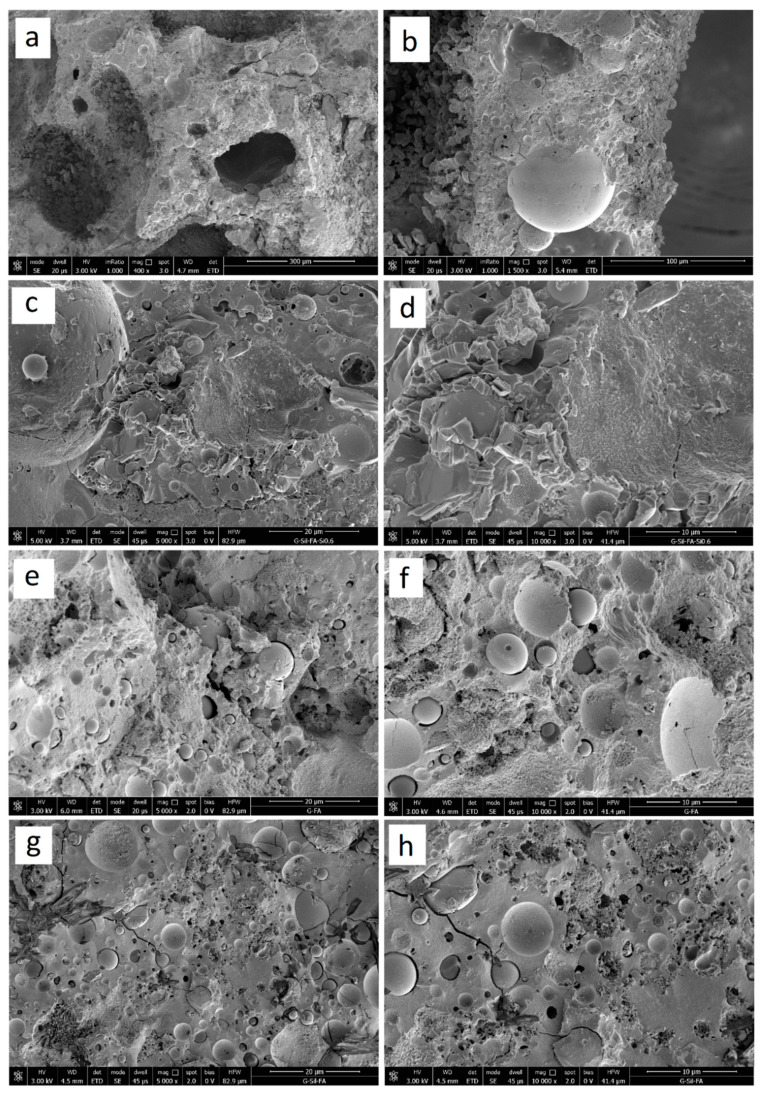
SEM micrographs of the hybrid geopolymer foam (GSyl06) at (**a**) 400, (**b**) 1500, (**c**) 5000, (**d**) 10,000 magnification; SEM micrographs of the unmodified geopolymer sample G at (**e**) 5000 and (**f**) 10,000 magnification; SEM micrographs of the GSyl sample at (**g**) 5000 and (**h**) 10,000 magnification.

**Figure 6 materials-13-02919-f006:**
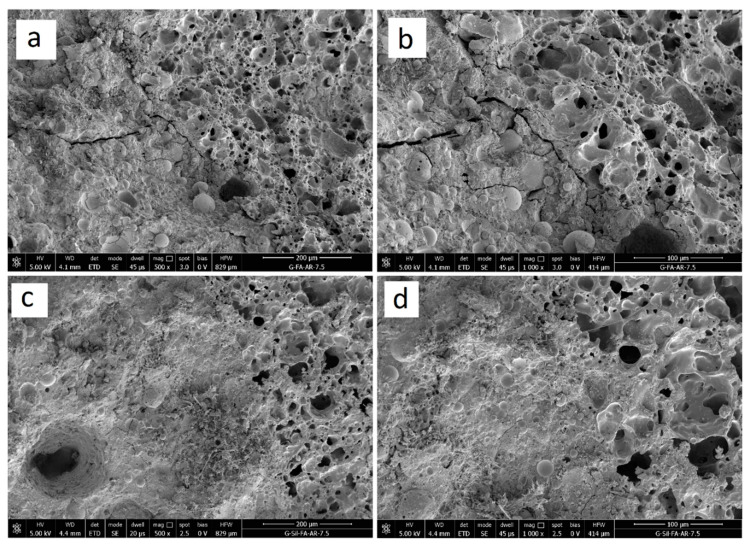
SEM micrographs at 500 (**a**) and at 1000 (**b**) magnification of the geopolymer with expanded clay (G–EC); SEM micrographs at 500 (**c**) and at 1000 (**d**) magnification of the hybrid geopolymer with expanded clay (GSyl–EC).

**Figure 7 materials-13-02919-f007:**
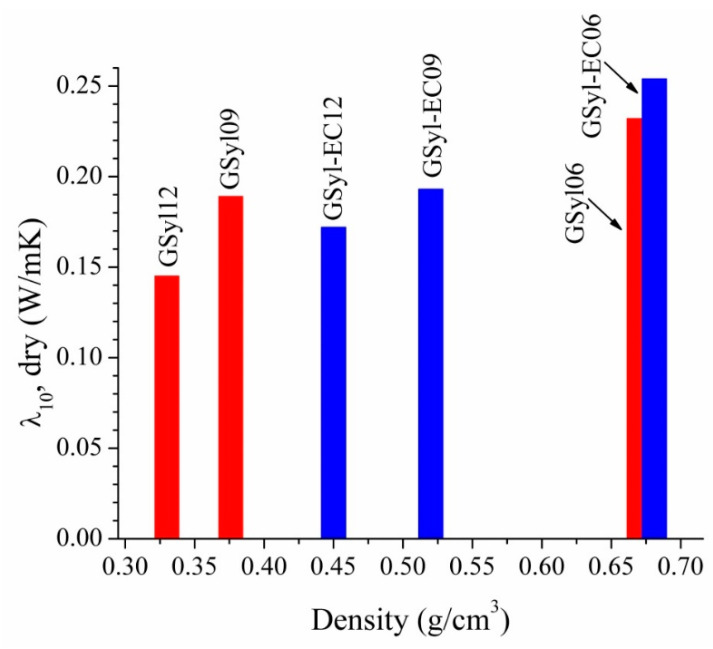
Thermal conductivity (W/m·K) for foamed GSyl (red bars) and GSyl–EC (blue bars) samples.

**Figure 8 materials-13-02919-f008:**
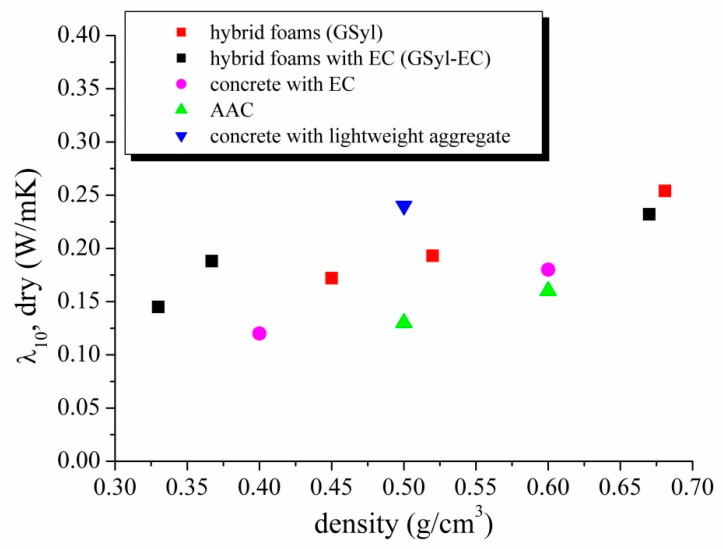
Thermal conductivity *vs* apparent density of the hybrid foams with (black squares) and without (red squares) EC and, for comparison, of lightweight concrete samples with EC (magenta circles); autoclave aerated concrete (green triangles); commercial sample of concrete with other lightweight aggregates, (blue triangles) [28] Table A.6, A.9, A.10.

**Figure 9 materials-13-02919-f009:**
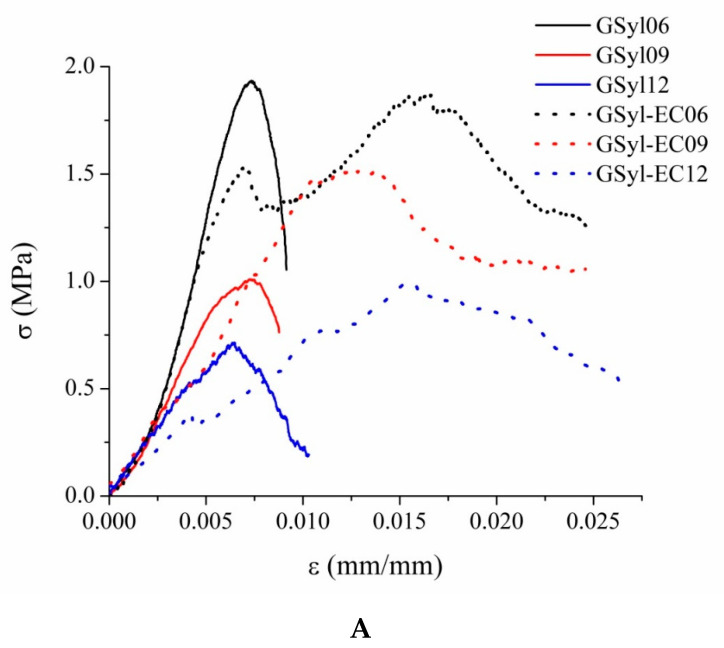
(**A**) Stress–strain curves in compression of foamed GSyl and GSyl–EC samples: GSyl06 (black solid curve), GSyl–EC06 (black dotted curve), GSyl09 (red solid curve), GSyl–EC09 (red dotted curve), GSyl12 (blue solid curve), GSyl–EC12 (blue dotted curve); (**B**) stress–strain curve in compression of foamed GSyl06ls hybrid foam obtained on a production scale (magenta curve) and, for comparison, stress–strain curve of GSyl06 (black curve), already reported in A and recorded on lab scale specimens. See text for details.

**Figure 10 materials-13-02919-f010:**
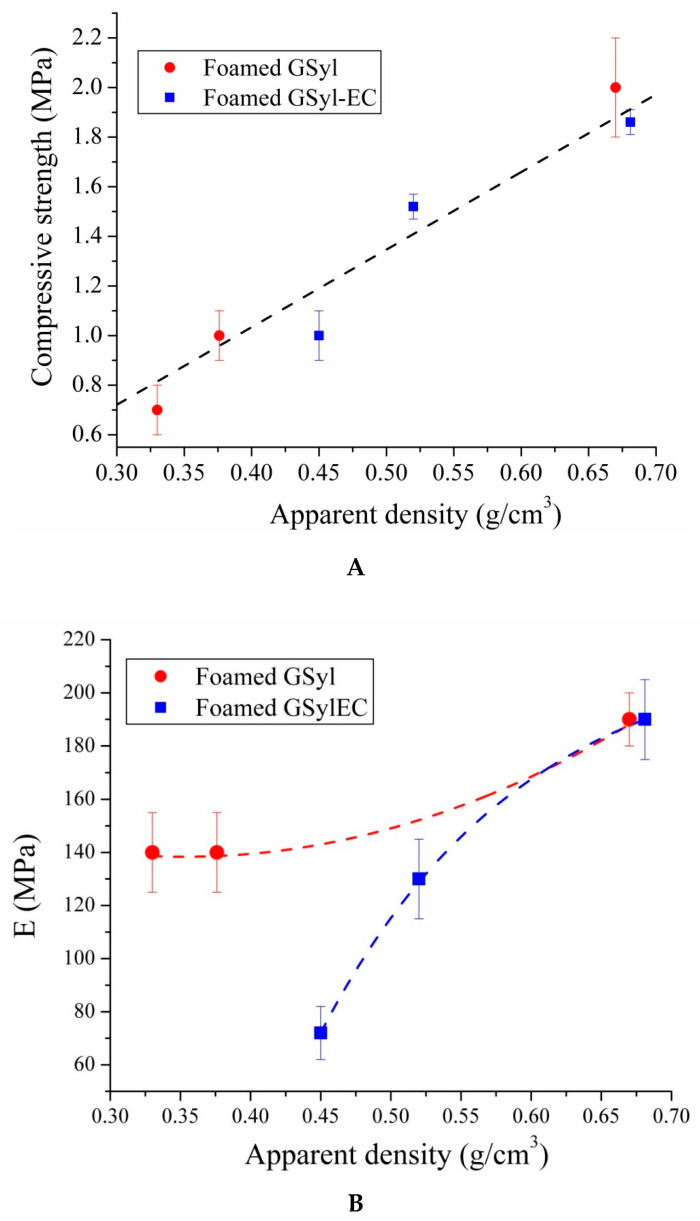
(**A**) Compressive strength and (**B**) Young’s modulus for foamed GSyl (red circles) and GSyl–EC (blue squares) samples as a function of apparent density. Lines are a guide for the eye.

**Figure 11 materials-13-02919-f011:**
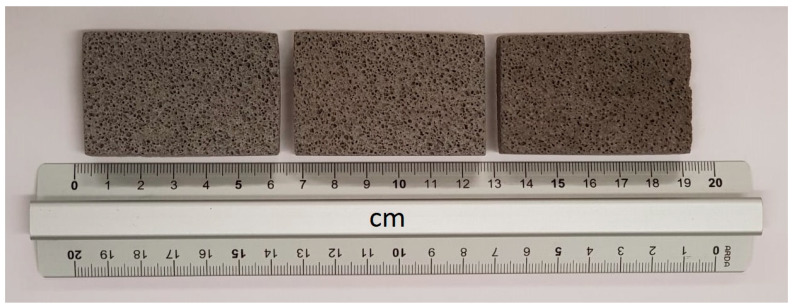
Optical images of polished section surfaces of GSyl06ls prepared by adding 0.06 wt % of Si^0^ to hybrid geopolymeric slurry and allowing the foaming process to occur in a 10-dm^3^ mold. The samples shown were obtained from the core section of one of the specimens prepared in production scale. The stress–strain curve collected for 50 × 50 × 50-mm^3^ specimens (obtained as described in the text) is shown in Figure 9B (in magenta).

**Table 1 materials-13-02919-t001:** Chemical composition (weight %) in terms of major oxides of the fly ash and sodium silicate solution used in this study. Other oxides (e.g., CaO, TiO_2_, SO_3_) with relative abundance less than 1 wt % are not indicated.

**Fly Ash**						
SiO_2_	Al_2_O_3_	Fe_2_O_3_	MgO	K_2_O	Na_2_O	Others
48.59	21.71	8.03	2.40	2.11	1.06	16.1
**Sodium Silicate Solution**						
SiO_2_			Na_2_O		H_2_O	
29.45			14.75		55.8	

**Table 2 materials-13-02919-t002:** Mix composition (wt %) and apparent density of the studied samples.

Sample	FA	SS	NaOH	EC	DMS	Si	Apparent Density (g/cm^3^)
**Solid Samples**							
G	66.17	24.46	9.37	–	–	–	1.662 ± 0.001
G–EC	61.20	22.53	8.67	7.5	–	–	1.562 ± 0.002
GSyl	59.55	22.01	8.44	–	10.0	–	1.471 ± 0.001
GSyl–EC	54.83	20.11	7.56	7.5	10.0	–	1.450 ± 0.003
**Foamed Samples**							
GSyl06	59.55	22.01	8.44	–	10.0	0.06	0.670 ± 0.002
GSyl09	59.55	22.01	8.44	–	10.0	0.09	0.376 ± 0.002
GSyl12	59.55	22.01	8.44	–	10.0	0.12	0.330 ± 0.001
GSyl–EC06	54.83	20.11	7.56	7.5	10.0	0.06	0.681 ± 0.003
GSyl–EC09	54.83	20.11	7.56	7.5	10.0	0.09	0.520 ± 0.002
GSyl–EC12	54.83	20.11	7.56	7.5	10.0	0.12	0.450 ± 0.003

FA = fly ash; SS = sodium silicate solution; EC = expanded clay; DMS = dimethylsiloxane oligomers; Si = silicon powder.

**Table 3 materials-13-02919-t003:** Averaged compressive strength (σ_c_) and strain (ε_c_), ultimate strength (σ_ult_), ultimate failure strain (ε_ult_) and Young’s modulus (E) of foamed GSyl and GSyl–EC samples as obtained from stress–strain curved reported in Figure 9.

Sample	σ_c_ (MPa)	ε_c_ (%)	σ_ult_ (MPa)	ε_ult_ (%)	E (MPa)
**GSyl06**	2.0 ± 0.2	0.7 ± 0.1	1.0 ± 0.1	1.0 ± 0.1	190 ± 10
**GSyl09**	1.0 ± 0.1	0.7 ± 0.1	0.8 ± 0.1	1.0 ± 0.2	140 ± 15
**GSyl12**	0.7 ± 0.1	0.7 ± 0.02	0.2 ± 0.1	1.0 ± 0.3	140 ± 15
**GSyl06ls**	5.3 ± 0.5	4.3 ± 0.1	4.5 ± 0.2	7.0 ± 0.3	170 ± 15
**GSyl–EC06**	1.86 ± 0.05	1.6 ± 0.1	1.2 ± 0.3	2.5 ± 0.2	190 ± 15
**GSyl–EC09**	1.52 ± 0.05	1.2 ± 0.1	1.0 ± 0.2	2.4 ± 0.1	130 ± 10
**GSyl–EC12**	1.0 ± 0.1	1.5 ± 0.2	0.5 ± 0.3	2.6 ± 0.3	72 ± 8

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
