# Peer review of "Hybrid Fly Ash-Based Geopolymeric Foams: Microstructural, Thermal and Mechanical Properties"

_materials, 2020, doi:10.3390/ma13132919_

Round 1
Reviewer 1 Report
- The characteristics of this new material should be compared with those of other materials in bibliography.
- The authors should recheck and correct the errors of typos and grammar before sending out this manuscript.
In my opinion, this article can be published in this journal if the authors offer enough new data. By all the aspects enumerated above, the author should make a minor amendment of the paper before its publication.
Author Response
Reviewer 1
- The characteristics of this new material should be compared with those of other materials in bibliography.
A comparison of the properties of our new materials with analogous commercial products or similar materials reported in scientific literature, is already reported in the text. For example:
- in par. 3.3, while discussing the thermal properties of the new material, Fig. 8 shows a comparison with lightweight concrete samples with EC, autoclave aerated concrete and commercial sample of concrete with other lightweight aggregates.
- as far as mechanical properties, par. 3.4 compares data obtained in this work with those already reported in literature for geopolymeric porous materials.
- The authors should recheck and correct the errors of typos and grammar before sending out this manuscript. In my opinion, this article can be published in this journal if the authors offer enough new data. By all the aspects enumerated above, the author should make a minor amendment of the paper before its publication.
Typos and grammar errors have been corrected. Several part of the text have been also rewritten to improve clarity.
Reviewer 2 Report
Dear Author,
A few specific comments are given below.
- The composition of the fly ash and sodium silicate are reported in table 1. How are they analysed?
- “The foaming process occurs thanks to the addition of Si 182 0 powder into freshly prepared 183 slurry: the formation of voids takes place due to the development of hydrogen gas.”
“On the contrary, thanks to higher viscosity of hybrid systems….”
Please rewrite the sentences.
- In line 202, its been mentioned that “the apparent densities”, what is that, how is the density determined/calculated?
- Line 295-303, the font size is different, please keep uniform font size
Overall the manuscript is not written clearly. The context is also not very clear and it could be accepted after rewriting.
Author Response
Reviewer 2
- The composition of the fly ash and sodium silicate are reported in table 1. How are they analysed?
Fly ash “EFA-Füller HP” was supplied by BauMineral GmbH (Herten, Germany). Its chemical composition was provided by the company by means of X-ray fluorescence spectroscopy. The sodium silicate solution was supplied by Prochin Italia S.r.l (Caserta, Italy). Also in this case, the chemical composition was provided by the company on the basis of the preparation protocol. The text of the manuscript has been changed accordingly.
- “The foaming process occurs thanks to the addition of Si (182) 0 powder into freshly prepared (183) slurry: the formation of voids takes place due to the development of hydrogen gas.” Please rewrite the sentences: “On the contrary, thanks to higher viscosity of hybrid systems….”
Sentences have been rewritten (lines 243-244).
- In line 202, its been mentioned that “the apparent densities”, what is that, how is the density determined/calculated?
In order to clarify this aspect, the following section has been added (lines 207-219).
“2.4 Apparent density determination
Apparent density measurements were carried out by means of an OHAUS-PA213 balance provided by Pioneer according to the following equation:
D=m_d/(m_s-m_i )
where md is the dry weight of the sample, ms is the weight of the soaked sample and mi is the weight of the soaked immersed sample. In order to determine their dry weight (md), the samples were dried at 110°C for 12 hours and weighed after cooling at room temperature. After that, the specimens were placed in an empty desiccator and kept in vacuum for 30 min. Later, the desiccator was filled with water and the samples were kept immersed for 2 h in vacuum and then weighed in order to obtain the weight of soaked sample (ms). The weight of the soaked immersed sample (mi) was then obtained by weighting the samples immersed in water at atmosphere pressure.)”
- Line 295-303, the font size is different, please keep uniform font size.
The font size was keep uniform.
- Overall the manuscript is not written clearly. The context is also not very clear and it could be accepted after rewriting.
The text has been improved by rewriting several sections.
Reviewer 3 Report
The article is well prepared and present a lot of results. The microscopy and photography images are very good. Some minor comments are suggested to improve the manuscript.
- Line 45: please add a reference to no flammability
- In conclusion, it would be good to recall the best mixes for thermal insulation.
- The abstract can also explain the best mix design for thermal insulation.
- Check figure 4 can be presented like figures 2, 7, 8, 9, 10
- Consider reducing the number of self-citations from the authors and to add other references.
Author Response
Reviewer 3
- Line 45: please add a reference to no flammability
- In conclusion, it would be good to recall the best mixes for thermal insulation.
- The abstract can also explain the best mix design for thermal insulation.
- Check figure 4 can be presented like figures 2, 7, 8, 9, 10
- Consider reducing the number of self-citations from the authors and to add other references.
All the suggested corrections have been taken into account. The manuscript has been modified accordingly.
Reviewer 4 Report
Manuscript Number: materials-827657
Title: Hybrid fly ash-based geopolymeric foams: microstructural, thermal and mechanical properties.
The paper is very interesting and it is very important for Materials. However, major corrections must be taking into account, as follow:
1.- Lines 37: Change the point after the reference bracket [1].
2.- Line 50: Change the point after the references bracket [2, 3].
3.- Line 53: Change the point after the references bracket [4, 5].
4.- Line 59: Change the point after the references bracket [6-8].
5.- Line 66: Change the point after the reference bracket [10]
6.- Line 70: Change the point after the references bracket [11-22].
7.- Line 73: Change the point after the references bracket [12, 18].
8.- Line 76: Change the point after the references bracket [18, 19].
9.- Line 79: Change the point after the reference bracket [19].
10.- Line 81: Change the point after the references bracket [18, 19].
11.- Line 95: Together words ((Caserta, Italy).Its).
12.- Line 135 (Table 2): some numbers and letters appear in bold (G, 66.17, 24.46, 9,37) in the rest do not. Why?.
13.- Line 182: Change the point after the reference bracket [19].
14.- Lines 295-303: Different size letters appear. Why?
15.- Line 303: Change the point after the reference bracket [26].
16.- Line 305: Change the point after the reference bracket [26].
17.- Lines 315-319: The name of the figure appears in abnormally large letters. Why?.
18.- Lines 390-391: Change the point after the references bracket [28, 29].
19.- References: Citations 2, 3, 4, 5, 9, 25, 26 and 27 are not cited according to journal regulations.
Best wishes

Author Response
Reviewer 4
The paper is very interesting and it is very important for Materials. However, major corrections must be taking into account, as follow:
1.- Lines 37: Change the point after the reference bracket [1].
2.- Line 50: Change the point after the references bracket [2, 3].
3.- Line 53: Change the point after the references bracket [4, 5].
4.- Line 59: Change the point after the references bracket [6-8].
5.- Line 66: Change the point after the reference bracket [10]
6.- Line 70: Change the point after the references bracket [11-22].
7.- Line 73: Change the point after the references bracket [12, 18].
8.- Line 76: Change the point after the references bracket [18, 19].
9.- Line 79: Change the point after the reference bracket [19].
10.- Line 81: Change the point after the references bracket [18, 19].
11.- Line 95: Together words ((Caserta, Italy).Its).
12.- Line 135 (Table 2): some numbers and letters appear in bold (G, 66.17, 24.46, 9,37) in the rest do not. Why?.
13.- Line 182: Change the point after the reference bracket [19].
14.- Lines 295-303: Different size letters appear. Why?
15.- Line 303: Change the point after the reference bracket [26].
16.- Line 305: Change the point after the reference bracket [26].
17.- Lines 315-319: The name of the figure appears in abnormally large letters. Why?.
18.- Lines 390-391: Change the point after the references bracket [28, 29].
19.- References: Citations 2, 3, 4, 5, 9, 25, 26 and 27 are not cited according to journal regulations.
All the suggested corrections have been taken into account. The manuscript has been modified accordingly.
Round 2
Reviewer 2 Report
Dear Authors,
Please check the formatting and grammar once again before submitting the final version.